# Genomic Characterization of a Bataï Orthobunyavirus, Previously Classified as Ilesha Virus, from Field-Caught Mosquitoes in Senegal, Bandia 1969

**DOI:** 10.3390/v16020261

**Published:** 2024-02-06

**Authors:** Cheikh Talibouya Toure, Idrissa Dieng, Safietou Sankhe, Mouhamed Kane, Moussa Dia, Moufid Mhamadi, Mignane Ndiaye, Ousmane Faye, Amadou Alpha Sall, Moussa Moise Diagne, Oumar Faye

**Affiliations:** 1Virology Department, Institut Pasteur de Dakar, 36 Avenue Pasteur, BP. 220, Dakar 12000, Senegal; cheikhtalibouya.toure04@gmail.com (C.T.T.); idrissa.dieng@pasteur.sn (I.D.); safietou.sankhe@pasteur.sn (S.S.); mouhamedkane2015@gmail.com (M.K.); moussa.dia@pasteur.sn (M.D.); moufid.mhamadi@pasteur.sn (M.M.); mignane.ndiaye@pasteur.sn (M.N.); ousmane.faye@pasteur.sn (O.F.); amadou.sall@pasteur.sn (A.A.S.); oumar.faye@pasteur.sn (O.F.); 2Department of Animal Biology, Faculty of Science, University Cheikh Anta Diop, BP. 5005, Dakar 10700, Senegal

**Keywords:** Bataï virus, Orthobunyavirus, Africa, Senegal, genome sequencing

## Abstract

Bataï virus (BATV), belonging to the Orthobunyavirus genus, is an emerging mosquito-borne virus with documented cases in Asia, Europe, and Africa. It causes various symptoms in humans and ruminants. Another related virus is Ilesha virus (ILEV), which causes a range of diseases in humans and is mainly found in African countries. This study aimed to genetically identify and characterize a BATV strain previously misclassified as ILEV in Senegal. The strain was reactivated and subjected to whole genome sequencing using an Illumina-based approach. Genetic analyses and phylogeny were performed to assess the evolutionary relationships. Genomic analyses revealed a close similarity between the Senegal strain and the BATV strains UgMP-6830 from Uganda. The genetic distances indicated high homology. Phylogenetic analysis confirmed the Senegal strain’s clustering with BATV. This study corrects the misclassification, confirming the presence of BATV in West Africa. This research represents the first evidence of BATV circulation in West Africa, underscoring the importance of genomic approaches in virus classification. Retrospective sequencing is crucial for reevaluating strains and identifying potential public health threats among neglected viruses.

## 1. Introduction

Bataï virus (BATV) is an emerging mosquito-borne virus that belongs to the genus *Orthobunyavirus* and the family *Peribunyaviridae* [1]. The virus was firstly isolated from Culex mosquitoes in Malaysia in 1955 [2] and since then has been detected in various regions of Asia and Europe, where it was also known as Calovo virus [3]. In Africa, one strain was isolated in 1967 in Uganda [4,5]. BATV has been found in several mosquito species and can infect humans and ruminants, causing fever, headache, joint pain, and neurological symptoms in the former [4,6].

The BATV genome consists of a tri-segmented, negative-sense, single-stranded RNA typical for Bunyaviruses. The S segment encodes the nucleocapsid (N) and the non-structural (NSs) proteins, the M segment encodes the virion surface glycoproteins (Gn, Gc) and non-structural proteins (NSm), and the L segment encodes for the replicase/transcriptase L protein [7].

Ilesha virus (ILEV) is another mosquito-borne virus belonging to the genus *Orthobunyavirus* [8]. ILEV was first isolated in 1961 from a 9-year-old girl who presented with fever and rash in Ilesha, a town in Western Nigeria [9]. Since then, it has been detected in several African countries and was detected in *Anopheles gambiae* [10,11]. ILEV can cause mild to severe disease in humans, ranging from febrile illness with exanthema to meningoencephalitis and hemorrhagic fever [11,12].

Here, we report on the genomic identification and characterization of a BATV strain in Senegal, previously classified as ILEV, using classical virological methods.

## 2. Materials and Methods

### 2.1. The Virus

This work was carried out as part of the characterization of *Peribunyaviridae* strains from the biobank of the WHO Collaborating Center for arboviruses and hemorrhagic fever viruses in the Institut Pasteur de Dakar (IPD). ArD 9870, a ILEV strain obtained from a mosquito, *Cellia gambiae s. I.*, collected on 9 October 1969 in Bandia, Senegal, and subsequently isolated on 17 June 1970 before storage in a freeze-dried form in the IPD biobank, was reactivated with 500 μL of 0.2% Bovine Serum Albumin (BSA) in PBS (1×).

### 2.2. Sequencing

First, RNA extraction was performed using the QIAamp viral RNA mini-kit (Qiagen, Hilden, Germany) following the manufacturer’s recommendations. Whole genome sequencing (WGS) was undertaken using a Illumina-based unbiased approach as previously described [13]. Briefly, a first step of host ribosomal RNA enzymatic depletion was carried out using specific probes and Oligo-dT as well as RNase H (New England Biolabs, Hitchin, UK). Depleted RNA was used as a template for first-stranded cDNA synthesis using the SuperScript IV Reverse Transcriptase kit (Invitrogen, Thermo Fisher, Waltham, MA, USA), followed by double-stranded cDNA synthesis with the Klenow exo-DNA polymerase (NEB, Hitchin, UK). Sequencing libraries were produced using the Nextera XT DNA Library Preparation kit (Illumina, San Diego, CA, USA) following the manufacturer’s recommendations. Genome assembly was carried out using the open-source metagenomics CZ-ID platform (http://czid.org, accessed on 15 March 2023) [14], with the defaults threshold filters applied for the reads of Quality Check, base-calling, and consensus generation. The sequencing metrics are summarized in Table 1.

### 2.3. Genetics Analyses and Phylogeny

The genomic segments (S, M, and L) were subjected to BLAST analysis against the GenBank database to identify the closest matching sequences. Furthermore, genetic distances were calculated using Mega software v10.1.8 [15] to evaluate the evolutionary relationships between the study strains and related sequences.

Phylogenetic analysis was carried out to elucidate the evolutionary relationships and clade formations. The sequences were aligned and curated using Bioedit version 7.2.6 [16]. Maximum Likelihood (ML) phylogenetic trees were constructed using Iqtree version 1.6.12 [17]. The robustness of tree topology was accessed with 1000 replicates and bootstrap values greater than 70% are shown on the branches of the consensus trees. The resulting trees were visualized using Figtree version 1.4.4 [18].

## 3. Results

### 3.1. Blast

BLAST analysis was performed on the genome of each segment obtained; closer nucleotide identities (98.28%, 96.59%, and 96.49% for the S, M, and L segments, respectively) were observed with the bataï strain UgMP-6830. However, the M segment grouped with the Ngari virus (NGAV) (strain Dakar D28542/4e (98.68% identity)).

### 3.2. Genetic Distance

The genetic distance between Ar D 9870 and the strain UgMP-6830 was evaluated, as well as some BATV, NGAV, and Bunyamwera (BUNV) strains. The sequences added were downloaded from Genbank (Appendix A), except for the “ArB218 Central African Republic 1968”, “ArMsam263 Kenya 1963”, “ArN31 Kenya 1974”, “ArY380 Cameroon 1971” and “ArYM52 Cameroon 1966” strains, which came from the IPD data bank. The analysis (Table 2) shows that, for all the M and L segments, the difference in the level of amino acids between the strain ArD 9870 from Senegal and strain UgMP-6830 was very low and even indicated a 100% homology for the S segment. It was, however, noted that, while the S and L segments of strain ArD 98 are more closely related to BATV, the M segment is more closely related to NGAV.

In addition, a genetic distance analysis was carried out between the BATV strains found in Africa, the strains found in Europe, and the strains found in Asia (Table 3). This shows that strains from Africa (AR D 9870 and UGMP-6830) were closer to the strains found in Asia.

### 3.3. Phylogeny

A phylogenic analysis was performed using the same dataset for genetic distance analysis with the addition of ILEV sequences downloaded from Genbank.

A total of 43 sequences were used for the phylogenic analysis of the S, M, and L segments. A sequence of Kairi virus was used as an outgroup (Appendix A). All three phylogenetic trees had four distinct clades, BUNV, BATV, ILEV, and NGAV. The strain Ar D 9870 clustered with BATV sequences for the S and L segments (Figure 1 and Figure 3), and with NGAV for the M segment (Figure 2).

## 4. Discussion

This work was carried out as part of the characterization of *Peribunyaviridae* strains from the biobank of the WHO Collaborating Center for arboviruses and hemorrhagic fever viruses in the Institut Pasteur de Dakar (IPD).

The blast analysis suggested that the Ar D 9870 strain was closer to the UgMP-6830 strain, a BATV strain isolated in Uganda in 1967 [5]. This result was confirm by the genetic distance analysis showing a very low difference at the level of amino acid and even a 100% homology with the S segment; this result is confirmed by Briese et al. [5]. However, the results show a close relation between the M segment of Ngari and the Ar D 9870 strain. That suggests a reassortment between Bataï and Ngari which results in the formation of new strains. This result was also found by Briese et al. [5] when they analyzed the UgMP-6830 strain, even if they classified it as Bataï. The M segment may play a major role in the virus cycle, affecting the interaction with the vector and also with the host [19,20]. Thus a reassortment might play a role in the virulence of the virus [21]. Genetic distance also shows a relation between African and Asian Strains and this same result was found by Mansfield et al. [6].

A phylogenic analyses was also performed during the study. As a member of the same serogroup, strains of BATV, NGAV, BUNV, and ILEV were used. The results confirmed the previous analysis [10] and suggested that Ar D 9870 and UgMP-6830 from Uganda could be contemporaneous BATV strains with simultaneous circulation in West and East Africa. This proves the important lack of information that has always existed regarding the real burden of the Bataï virus and highlights the necessity of further studying the seroprevalence among the population and prevalence among vectors.

## 5. Conclusions

To the best of our knowledge, this work describes the first report of BATV circulation in West Africa. Indeed, phenotypic-based virological characterization methods, such as the complement fixation test, have limitations regarding strains belonging to the same serogroup with no specific antibody reactions [16,17]. WGS allowed to reclassify a strain isolated in Senegal in 1969 as BATV, which was mistakenly identified as ILEV using the complement fixation test. Retrospective sequencing will be relevant for both BATV and ILEV strains, but also for other neglected viruses collected and previously classified using non-molecular taxonomic approaches in order to refine classification and enable the identification of potential pathogens of public health concern.

## Figures and Tables

**Figure 1 viruses-16-00261-f001:**
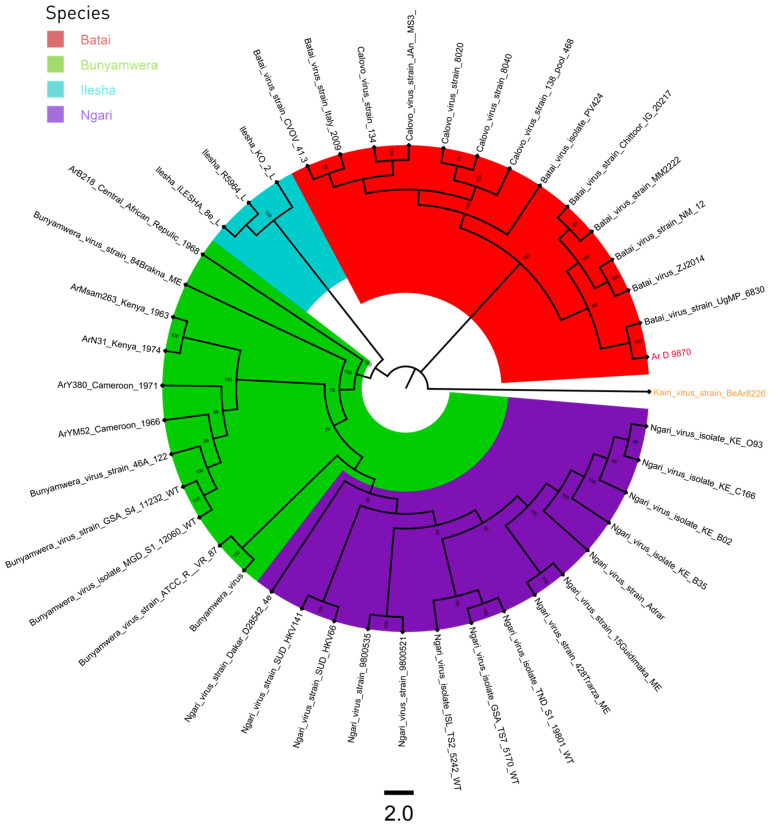
L segment phylogenetic tree of Bunyamwera, Bataï, Ilesha, and Ngari viruses. The Kairi virus, colored in orange, was used as an outgroup, and the sequence Ar D 9870, colored in red, was that previously classified as the Ilesha strain.

**Figure 2 viruses-16-00261-f002:**
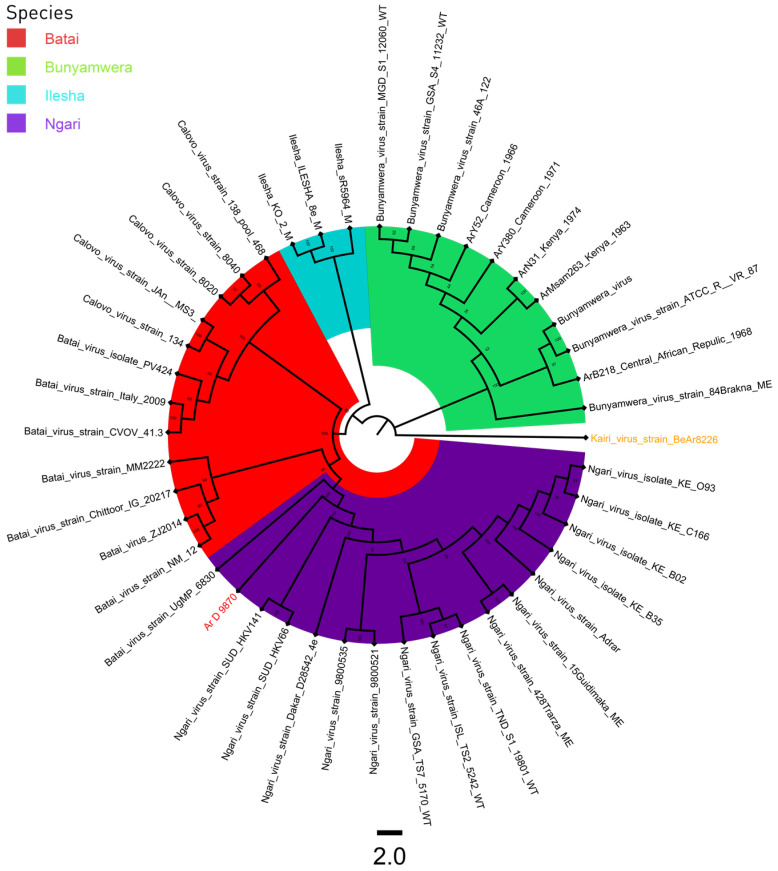
M segment phylogenetic tree of Bunyamwera, Bataï, Ilesha, and Ngari viruses. The Kairi virus, colored in orange, was used as an outgroup, and the sequence Ar D 9870, colored in red, was that previously classified as the Ilesha strain.

**Figure 3 viruses-16-00261-f003:**
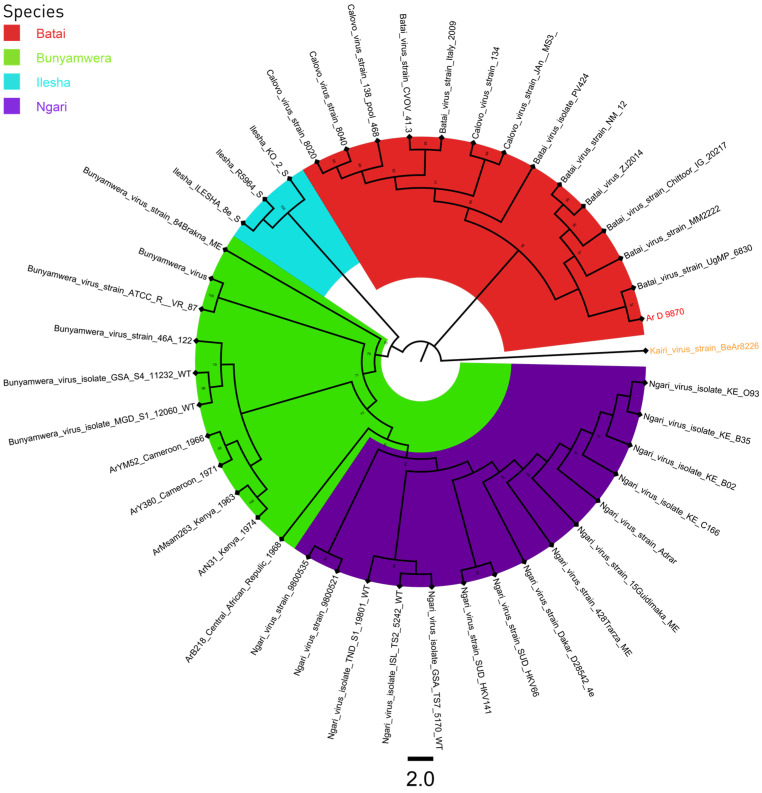
S segment phylogenetic tree of Bunyamwera, Bataï, Ilesha, and Ngari viruses. The Kairi virus, colored in orange, was used as an outgroup, and the sequence Ar D 9870, colored in red, was that previously classified as the Ilesha strain.

**Table 1 viruses-16-00261-t001:** Sequencing metrics summary.

	Segment L	Segment M	Segment S
Mapped Reads	23,210	17,746	1667
GC Content	33.9%	36.4%	40.6%
SNPs	241	148	14
%id	96.5%	96.6%	98.5%
Informative Nucleotides	6870	4338	927
% Genome Called	100%	98.5%	98.1%
Missing Bases	0	63	16
Ambiguous Bases	0	0	0
Reference length	6870	4404	945
Coverage Depth	403.9×	477.5×	191.7×
Coverage Breadth	100%	99.9%	100%
Reference Sequence	JX846603.1-Bataï virus strain UgMP-6830 segment L, complete sequence	DQ436460.1-Bataï virus strain UgMP-6830 segment M polyprotein gene, complete cds	JX846601.1-Bataï virus strain UgMP-6830 segment S, complete sequence

**Table 2 viruses-16-00261-t002:** Genetic distance between groups. The number of amino acid differences per sequence from averaging over all sequence pairs between groups are shown. Standard error estimate(s) are shown above the diagonal in blue. In italics: the closest group to Ar D 9870; in bold: the second closest group.

	AR D 9870	Bataï	UgMP-6830	Bunv	Ngari
S segment					
AR D 9870		1.375	0.000	3.887	3.797
BATAÏ	**3.167**		1.375	3.833	3.759
UGMP-6830	*0.000*	3.167		3.887	3.797
BUNV	16.909	17.348	16.909		0.599
NGARI	16.267	16.767	16.267	1.170	
M segment					
AR D 9870		8.539	5.073	16.775	4.313
BATAÏ	125.667		8.151	16.128	8.384
UGMP-6830	*25.000*	117.667		16.734	5.471
BUNV	484.909	486.159	485.273		16.751
NGARI	**30.733**	131.533	42.467	492.103	
L segment					
AR D 9870		11.491	7.586	20.035	20.040
BATAÏ	**221.000**		11.907	19.265	19.324
UGMP-6830	*59.000*	233.250		19.934	19.856
BUNV	609.636	611.689	610.727		6.607
NGARI	608.667	614.711	608.333	91.867	

**Table 3 viruses-16-00261-t003:** Genetic distance between groups. The number of amino acid differences per sequence from averaging over all sequence pairs between groups are shown. Standard error estimate(s) are shown above the diagonal in blue. In italics: the group closer to Bataï Africa.

	Bataï_Africa	Bataï_Asia	Bataï_Europe
S segment			
Bataï_Africa		1.030	2.001
Bataï_Asia	*1.250*		2.285
Bataï_Europe	4.143	5.393	
M segment			
Bataï_Africa		8.587	11.173
Bataï_Asia	*103.375*		10.986
Bataï_Europe	144.214	148.679	
L segment			
Bataï_Africa		11.046	14.762
Bataï_Asia	*173.750*		14.726
Bataï_Europe	275.571	290.536	

## Data Availability

Data are contained within the article and Appendix A.

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
