# Peer review of "Genomic Characterization of a Bataï Orthobunyavirus, Previously Classified as Ilesha Virus, from Field-Caught Mosquitoes in Senegal, Bandia 1969"

_viruses, 2024, doi:10.3390/v16020261_

Round 1

Reviewer 1 Report

Comments and Suggestions for Authors

Comments on Toure et al., Genomic characterization of a Batai Orthobunyavirus, previously classified as Ilesha virus, from field-caught mosquitoes in Senegal, Bandia 1969.

This is a solid characterization of a previously misclassified virus, originally identified as an Ilesha virus, using deep sequencing and phylogenetic analysis.  The virus was retrieved and reactivated from a 1970 cyrobanked stock isolated originally from a wild-caught Cellia gambiae mosquito in Senegal.

The data clearly indicate that the virus is more closely related to the Batia viruses than the Ilesha viruses, and so should be formally reclassified.  The paper is scientifically sound and the finding is of general interest.  In my opinion, the paper could be strengthened and made of further general interest if more discussion were presented as to the significance of the finding that Batai-like viruses were simultaneously circulating in both East and West Africa, as this appears to be a new observation.  Also, because the M segment of this virus appears significantly more closely related to the Ngari viruses than either the Batai or the Ilesha viruses, some discussion should be given to the idea that this particular strain is derived from a reassortment event between Batai and Ngari viruses.  It may also be worth discussing that since NSm may be important for interactions of the virus with its mosquito host, a putative reassortment event may have something to do with the isolation from a particular mosquito species.

Minor points:

Table 1: Appears to be an error in the % genome called column for L segment

Line 97 typo: “expect” should be “except”

Line 99 typo: “coming” should be “came”

Author Response

Dear Reviewer,

We are grateful for the time spent for the review of this work.

We tried to respond to all the points raised (please confer to the associated file) and we then hope that the new version will fit better wiith your expectations.

Thank you again for your valuable review.

Best regards,

The authors

Reviewer 2 Report

Comments and Suggestions for Authors

This paper retroactively investigates a strain of Ilesha virus in a biorepository to confirm identity. Interestingly, the results indicate that previously classified Ilesha virus was in fact Batai virus, which is underreported in Africa. While interesting, this result seems more like a genome announcement rather than an entire paper, but I defer to editors. There is perhaps more information in the phylogenetic analysis that could be expounded on to justify an entire research article. However, there are some missing information as there are some Bunyamwera strains that have been isolated from Rwanda and deposited into GenBank (see Dutuze et al 2020), as well as possible Batai segments. It is critical that these be included as BATV circulation in Africa as a whole is underreported and this may affect conclusions.

As the phylogeny of this single viral isolate is the crux of this paper, some more information about how comparison genomes were chosen should be included. In GenBank, there are 169 at least partial hits for BATV. What was the exclusion criteria for not including all (as assumed based on Supplemental Table and Figure 1-4).

Throughout, the authors make note that the M segment clusters more with NRIV. While this is not incorrect, it should be discussed that NRIV is the reassortant of BUNV and BATV and the M segment was inherited from BATV initially.

Minor comments:

In Table 1, is the L segment 1% genome called? I’m assuming, but since the notation is different, was unsure. Make sure notation is consistent throughout.

Line 79: Genetic should not be capitalized.

Table 3 is hard to understand. It isn’t clear where the standard errors are – even with the caption. Consider simplifying.

Author Response

Dear Reviewer,

We are grateful for the time spent for the review of this work.

Your different comments will definitely contribute to improve the quality of the manuscript.

We then tried to respond to all the points raised by you (please confer to the associated file) and we hope that the new version will fit better with your expectations.

Thank you again.

Best regards,

The authors
